# Ram Semen Cryopreservation for Portuguese Native Breeds: Season and Breed Effects on Semen Quality Variation

**DOI:** 10.3390/ani13040579

**Published:** 2023-02-07

**Authors:** João Pedro Barbas, Jorge Pimenta, Maria Conceição Baptista, Carla Cruz Marques, Rosa Maria Lino Neto Pereira, Nuno Carolino, João Simões

**Affiliations:** 1Department of Biotechnology and Genetic Resources of Instituto Nacional de Investigação Agrária e Veterinária, Quinta da Fonte Boa, 2005-048 Vale de Santarém, Portugal; 2CIISA-AL4AnimalS-Faculty of Veterinary Medicina, Universidade de Lisboa, 1300-477 Lisboa, Portugal; 3Department of Veterinary Sciences, Veterinary and Animal Research Centre (CECAV), AL4AnimalS, School of Agricultural and Veterinary Sciences, University of Trás-os-Montes and Alto Douro (UTAD), 5000-801 Vila Real, Portugal

**Keywords:** semen cryopreservation, ram, spermatozoa, local breeds, germplasm bank, reproductive biotechnologies

## Abstract

**Simple Summary:**

This study mainly aimed to evaluate the seasonal and breed effects on spermatozoa (SPZ) traits of ten Portuguese sheep breeds, cryopreserved and stored in the Portuguese National Germplasm Bank. The mean volume and SPZ concentration of the ejaculate was 0.77 mL and 5039 ± 51 × 10^6^/mL, respectively, and were affected by breed but not by season. The viability (alive SPZ), individual motility, and abnormal morphology of fresh semen were 63.2 ± 0.2%, 61.41 ± 0.2%, and 10.1 ± 0.1%, respectively. A significant decrease of 28.5% of the viability and 35.5% of individual motility, as well as an increase of 54.5% in the SPZ abnormal morphology was observed in thawed semen. An effect of breed, but not of season, was observed on these SPZ traits. Nonetheless, breed × season interactions influenced individual motility and abnormal morphology, except for tail defects. Overall, the effect of season on ejaculate and SPZ traits was not significant and local rams are able to be SPZ donors for cryopreservation during the whole year in our latitudes.

**Abstract:**

The semen quality is one of the determinant factors of ram semen cryopreservation. The present retrospective study aimed to characterize the seasonal ram pattern during the year for ten Portuguese local sheep breeds, hypothesizing that the breed and season had low effects on the main spermatozoa traits. A total of 1471 ejaculates were used and evaluated (fresh semen) from 85 rams between 2004 and 2020 and re-evaluated after thawing (thawed semen). The effect of breed, season, and sperm cryopreservation on nine semen traits were evaluated. The volume per ejaculate, spermatozoa (SPZ) concentration, and total number of SPZ per ejaculate, were affected by breed (*p* < 0.001) but not by season (*p* > 0.05). As expected, the semen processing was the most significant (*p* < 0.001) factor of variation on seminal parameters. Moreover, breed and interactions between breed × semen processing, modulated the response of alive SPZ, abnormal morphology, head, and intermediate piece defects. In fresh semen, season only affected the intermediate piece defects due to the highest percentage observed between February and April period in some breeds. Overall, and despite the mentioned particularities, there were similarities among the ten local breeds. We also concluded that the seasonal effect on ejaculate and SPZ traits is not significant in our region. These local ram breeds have low seasonality and can be employed in natural mating as well as semen donors for cryopreservation and assisted reproductive biotechnologies during the whole year at our latitude.

## 1. Introduction

Portugal is considered a hotspot of animal biodiversity including 16 native ram (sheep) breeds [1]. However, extreme situations persist, with some autochthonous breeds of small ruminants still considered as threatened (<6500 individuals) or rare (<3500 individuals), according to the classification based on the existing number of females exploited in purebred lines. Over the last 20 years, there has been a significant decrease (−30%) in ewes exploited in purebred lines [2]. Most native breeds are exploited for dual purposes, meat and milk, and in extensive production systems, i.e., in mountain and/or pasture areas. The selection of indigenous breeds and grazing techniques adapted to different geographical and climatic realities make extensive grazing a cultural heritage to be preserved, an activity with a great capacity to adapt to climate change, and a useful contribution to the settling of young people in inland regions.

The local breeds, at our latitudes, have low seasonality and can show reproductive activity throughout the entire year [3], similar to local breeds from low latitudes [4,5]. However, previous studies with Serra da Estrela and Saloia ewe breeds, have shown that during winter till mid spring (January till mid-April) 20–25% of these animals showed silent estrus, anovulatory, or irregular estrous cycles, which are influenced by breed, age, parity, body score, nutrition, and production system [2,5,6]. In addition, males from the Serra da Estrela and Merino breeds exhibited libido variation and semen quality (fresh and frozen semen) according to individual and season of semen collection [7,8]. These studies suggested that natural mating is possible throughout the year without significant decreases in flock fertility compared to when one annual lambing is considered (which is the usual extensive system) [9]. Furthermore, a similar profile was observed using artificial insemination (AI) schedules with refrigerated semen collected in autumn and spring [9,10]. Moreover, it is well known that the stimulation of sexual activity in inactive rams starts about 1 to 1.5 months (spermatogenesis cycle = 45 days) earlier than ewes [5]. Nonetheless, the real impact of the seasonal effect of semen collection on semen cryopreservation has not been consistently evaluated in our native breeds [2].

Research has been done in other countries in several latitudes and climates, focused on the influence of environmental, breed, individual, semen extenders and processing methods upon quantitative and qualitative seminal parameters in fresh and frozen semen (e.g., [11,12,13]). Globally, some variation in reproductive traits has been observed in native European breeds, which has been attributed to intrinsic characteristics, breeding systems and environmental conditions, highlighting the rainfall regime and summer temperatures. These details are mandatory to well characterize each local breed per se, mainly for semen cryopreservation and its future use.

In the National Zootechnical Station (EZN, Portugal), the first experiments on ram semen cryopreservation using animals of native breeds (Serra da Estrela, Merino Branco and Saloia) were performed in 1998 [2,9]. After this date and until now, several other local breeds have been evaluated and their semen incorporated into the Portuguese Animal Germplasm Bank (BPGA) for ex situ conservation purposes [14]. An increase in the number and size of germplasm banks to face the significant decrease of biodiversity due to the rise of endangered and or extinction of domestic breeds, has been reported in the last 10 years [14].

The main aims of this retrospective study were to (1) describe the values of spermatozoa (SPZ) traits in fresh and frozen semen collected and evaluated at EZN, from 2004 to 2020 in ten Portuguese ram breeds from different flocks, and (2) evaluate the breed and seasonal effect of semen collection periods on seminal parameters in fresh semen and after the frozen-thawing cycle. We hypothesize that these local ram breeds can serve as semen donors to SPZ cryopreservation without significant seasonal variations in seminal parameters, both in fresh and frozen semen.

## 2. Materials and Methods

This study was approved by INIAV, one of the promotors of the Portuguese Animal Germplasm Bank.

### 2.1. Animals and Management

Ten native Portuguese ram (sheep) breeds accounting for 85 animals, with a mean age of 36.3 months (95% IC: 35.0–37.6 months), were used in the present study and housed at EZN (lat: 39°11′57.3″, long: 8°44′22.5″), between 2004 and 2020 throughout the year. Rams of each breed were managed to maintain an adequate body condition score. At nutritional level, the rams were grouped in three collective parks, according to the years, with ad libitum hay and a commercial feed concentrate (1 kg per day). Rams were dewormed and vaccinated, twice a year, against clostridial and pasteurellosis diseases and screened for brucellosis. The flock was classified as brucellosis-free for each year. These rams were involved in specific projects, namely Portuguese Animal Germplasm Bank (BPGA) throughout these years and were dismissed when each animal had 250 doses of frozen semen (which are the requirements of BPGA to complete ram ejaculates donation) of good quality, namely ≥30% in individual motility (IM) and live SPZ and ≤20% abnormal SPZ per dose.

### 2.2. Semen Collection, Evaluation and Processing

For each ram, 1–2 ejaculates were collected per week over several months. Semen was collected from 85 rams by the artificial vagina method using an ewe in estrus. The number of sessions were different among rams, being determined by fresh and frozen semen quality as defined by quantitative and qualitative seminal parameters [6] (also, see above) and by the number of doses obtained per ejaculate. Each male should provide a total of 250 doses. After collection, semen was immediately maintained in a water bath at 30 °C. Additionally, ejaculates with reduced volume (<0.4 mL) and/or poor quality (IM < 55%, SPZ concentration < 2000 × 10^6^ SPZ cells per mL) were rejected. Overall, a total of 1471 ejaculates from 10 breeds were collected, evaluated, and validated for quantitative and qualitative seminal parameters as reported in Table 1. Only ejaculates with previously defined required standards were used in this study.

The semen was evaluated according to the methods described by [15]. Semen volume and concentration were immediately measured with a graduated collection vial and by a calibrated spectrophotometry (WPA-S106), respectively [7]. To evaluate SPZ concentration, 10 µL of raw semen from each ejaculate were placed in a cuvette and diluted in 3990 µL of physiologic serum (dilution rate 1:400). Afterwards, the SPZ concentration was calculated as the number of SPZ cells per mL (×10^6^/mL) using a wavelength of 550 nm (nanometer). The total number of SPZ were determined by multiplying semen volume by concentration. The IM (%) from each ejaculate was determined subjectively using an aliquot of 10 µL of diluted semen (1:10) in saline physiologic serum (200× magnification using a phase-contrast microscope) which was placed and covered with a coverslip on a warm slide (37 °C) as described by [13]. The SPZ viability (alive; %) and SPZ morphology, i.e., abnormal morphology (%), head defects (%), middle piece defects (MP; %) and tail defects (%) were evaluated in a smear of diluted semen stained with nigrosine-eosin (magnification of 1000× using an immersion oil objective; Olympus BX40 microscope^®^, Tokyo, Japan) in accordance with [13]. Briefly, one hundred SPZ cells were counted using a hemocytometer. Live SPZ were stained white or pink. Dead SPZ were stained red or purple. This smear was also used to determine abnormal SPZ cells according to the method described by [15].

For semen cryopreservation, a semen extender that were formulated in our Andrology laboratory placed at EZN and named EZN S-EXT [16] was used.

The composition was 15% of EY (egg yolk); 6% of glycerol; 2.1805 g of Tris; 0.3 g of Glucose; 1.194 g of citric acid; 0.05 g of Penicillin and 38 mL of sterile bi-distilled water. The EY was obtained from daily fresh chicken eggs. All chemicals were bought from Sigma-Aldrich (Steinheim, Germany). The EY was manually separated from albumen using a filter paper and a sterile syringe to pierce the chalaza.

The SPZ ejaculates were diluted to give a final SPZ concentration of 800 × 10^6^ SPZ cells/mL. Afterwards, diluted semen was packed in Cassou Straws of 0.25 mL (IMV Technologies, l’Aigle, France) and sealed with polyvinylpyrrolidone powder. Straws of different colors were identified with the identification number of the ram, breed, semen collection center and date of semen collection. Filled straws were placed in a glass tube using a vial with water at 28 °C and then cooled to 4 °C for 4 h (equilibration period) in a refrigeration chamber set to 4 °C. Then, straws were placed horizontally in a float freezing rack (Minitübe GmbH, Tiefenbach, Germany), 4 cm above the liquid nitrogen level in a stainless box covered with Styrofoam. Straws were frozen in nitrogen vapors at −120 °C during 20 min. Afterwards, they were removed and plunged in liquid nitrogen at −196 °C.

About one week after freezing, the straws were thawed in a water bath at 38 °C for 1 min. The content of each straw was diluted in 1 mL of saline solution and homogenized for 90 s. After 2 min, semen aliquots (10 µL) were used to evaluate motility, viability, and morphological defects as previously described by [16].

### 2.3. Breeding and Non-Breeding Season

Semen collection with an artificial vagina was done throughout the year till each animal had produced 250 doses of frozen semen of good quality, i.e., according to the previously described requirements of BPGA. In this work, two periods of semen collection, namely the breeding season and the non-breeding season, were considered.

The breeding season was defined as from September to January (mainly fall and early winter seasons; period 1) as defined for high (about >45 N) latitudes [5,17]. The non-breeding season [18] was subdivided into two periods: from February to April (period 2), representing a short nonbreeding season for sheep local breeds, in which a superficial seasonal anestrous is detected, such as described for latitudes about 39–42 N, [18] and May to August (period 3), which may be considered the transition period for the breeding season where local ewe breeds [2,7] and goats [19,20,21] can be naturally mated.

### 2.4. Statistical Analysis

An arcsine square root transformation (Sqrt) of all ejaculate and SPZ traits was made to reach or approach normal distribution previously evaluated by Shapiro–Wilk W test.

A multivariable mixed linear model for repeated measures, using the restricted maximum likelihood (REML) method, was built for each reproductive trait (Alive SPZ, IM and morphology traits) following the equation:Y_ijmo_ = H_i_ + L_j_ + B_o_ + (H × L)_ij_ + (H × B)_io_ + (L × B)_oj_ + (H × L × B)_ijo_ + t_mi_ + e_ijmo_
where,

Y_ijmo_ is a vector of all observations and represented by the least square value;

H_i_ is the fixed effect for breed (10 levels considering all the breeds);

L_j_ is the fixed effect for season (3 levels: September and January, February to April, and May to August);

B_o_ is the fixed effect for semen processing (2 levels: fresh and thawed semen);

(H × L)_ij_, (H × B)_io_ and (L × B)_jo_ are the two-way interactions;

(H × L × B)_ijo_ is the three-way interaction;

T_mi_ is the random effect for animal (_m_) within the breed; and,

e_ijmo_ is a vector of residuals.

For ejaculate volume, SPZ concentration, and total number of SPZ, only fresh semen was used. The fixed effect for semen processing, as well as the three-way interaction, were removed from the general equation.

The Tukey test was used to test differences of fixed factors and the Wald test to estimate variance component of the mixed factor.

The software JMP^®^ 14 for Windows (SAS Institute, Cary, NC, USA) was used to build the models. Except for the descriptive analysis (Table 1), all the results were presented as least squares means ± (Sqrt) SEM or 95% confidence interval (95% CI) for a 0.05 level of significance.

## 3. Results

### 3.1. Ejaculate Traits (Volume, Spermatozoa Concentration and Total Sperm)

For these years, considering 10 local Portuguese ram breeds and 1471 ejaculates from 85 rams, the least square mean volume per ejaculate, was 0.77 mL (95% CI: 0.75–0.78) and SPZ concentration was 5039 ± 51 × 10^6^/mL (95% CI: 4940–5138 × 10^6^/mL). The total SPZ count per ejaculate was 3895 ± 58 × 10^6^ (95% CI: 3780–4009 × 10^6^; *n* = 1471) (Table 2).

All three parameters were affected by breed (*p* < 0.001; Table 3), but a breed × season interaction was observed for SPZ concentration (*p* < 0.01) and total SPZ (*p* < 0.05). The breed × season interaction effect on the SPZ concentration was due to the highest value of the Churro Galego Mirandês (6136 ± 126 × 10^6^ SPZ/mL) during period 3, contrasting with the lowest value for Churro Galego Bragançano (3780 ± 142 × 10^6^ SPZ/mL) observed during periods 3 and 2 (3678 ± 140 × 10^6^ SPZ/mL). Furthermore, the highest SPZ concentration for the Churro Galego Mirandês during period 3 contrasted with the Churro do Campo breed during period 1 (3537 ± 155 × 10^6^ SPZ/mL; the highest value for this breed). The breed × season interaction effect on total SPZ was due to the Churro do Campo breed, where there was detected an inverse pattern during period 2 (highest values between February and April, 1962 × 10^6^ SPZ), relative to the other breeds.

A significant effect (*p* < 0.001) of the animal on these three studied traits was observed in all the breeds. The RELM variance component estimate of the animal variable represented 18.5% (0.005), 32.7% (0.06), and 26.3% (0.08) of the total variance (*p* < 0.001) for volume, SPZ concentration, and total SPZ count, respectively. The remaining variance of each reproductive trait concerned the residual random effect.

### 3.2. Spermatozoa Viability, Motility and Morphology

Overall, in fresh semen, the SPZ viability (alive SPZ) was 63.2 ± 0.2%, IM 61.41 ± 0.2%, abnormal morphology 10.1 ± 0.1%, head defects (head D) 4.2 ± 0.1%, IP defects (IPD) 1.8 ± 0.1%, and tail defects (tail D) 2.3 ± 0.1%.

According to Table 4, significant effects of breed (except for tail D) and semen processing were observed for all the SPZ traits. The semen processing was the most significant effect.

The magnitude of the LSmean differences (*p* < 0.001) of the SPZ traits between fresh and thawed semen for these local ram breeds are reported in Figure 1. After thawing, the percentage of alive SPZ and of their IM decreased. The increased percentage of abnormal SPZ morphology was due to the rise of head D and tail D. Inversely, the IPD decreased after thawing (Figure 1e).

In fresh semen, the breed effect on four SPZ traits, namely SPZ alive, abnormal morphology, head D, and IPD), was observed (Table 5).

In thawed semen, the breed only affected SPZ abnormal morphology and head D (Table 6). In fact, the most significant interactions were related to the breed (breed × thawing and breed × season interactions), which are reported as Appendix A.

In fresh semen, the season effect (*p* < 0.001) and breed × season interaction was observed in % IPD (*p* < 0.001 (Figure 2). It was due to the highest percentage of SPZ IPD during period 2. This difference was not observed (*p* > 0.05) in the corresponding thawed semen.

A significant animal random effect of Animal (*p* < 0.01) on all these six SPZ traits was observed in all the breeds. The RELM variance component estimate of animal variable represented 50.3% (1.48), 66.9% (0.54), 36.4% (0.44), 38.1% (0.46), 6.3% (0.05), and 10.8% (0.10) of the total estimate variance of SPZ alive, IM, abnormal morphology, head, IP, and tail defects, respectively.

## 4. Discussion

The present study highlighted the effect of breed (genotype) and freeze-thawing cycle for 10 Portuguese ram breeds, as well as the limited effect of the season of semen collection, on ejaculate and SPZ traits. In native breeds, besides photoperiod, other non-exclusive factors such as body condition score and feed availability [6], which were controlled in this study, and the air temperature, needs to be considered in a sheep extensive operating system. Differences for SPZ traits in our local breeds were observed regarding the season (breed × season interaction) and the freeze-thawing cycle (breed × semen processing interaction), and as previously reported [7,8,10].

The LSmean ejaculate volume and SPZ concentration was 0.77 mL and 5039 ± 0.051 × 10^6^/mL, respectively, varying among breeds and without significant seasonal effect. Contrarily to the ejaculate volume of Merino rams (average = 1.26 mL; [22]; Spain), a large breed, and Dorper (1.2 mL), Poll Dorset (1.1 mL), White Suffolk (1.1 mL), or Australian Dohne Merino rams [23], none of the Portuguese breeds reached more than 0.90 mL per ejaculate. Nonetheless, the SPZ concentration of our local ram breeds were higher than the Merino breed (average = 4205 × 10^6^/mL; [22]). Consequently, the total number of SPZ per ejaculate remains statically similar to our Portuguese rams (3895 ± 58 × 10^6^), except for the Churro do Campo rams, which presented the lowest value (1764 ± 169 × 10^6^) and an average volume of 0.52 mL. This endangered breed also presented the highest value of total SPZ per ejaculate during the period 2 (February and April; seasonal anestrus), which was responsible for the breed × season interaction for this trait. In 2004, this breed was considered extinct [24]; and in 2019 (last census), only 35 rams were recorded in the pedigree book of this breed [25]. Previous studies showed that native breeds had low seasonality being possible reproduction throughout the year, when one annual lambing year were selected, which is usual in extensive systems [7,8]. However, during winter and early spring, reproductive efficiency was lower. Nonetheless, in native breeds exploited in intensive systems, individual and seasonal variations were more easily displayed [26]. Spermatogenesis and semen quality is known to be influenced by several factors, including age, nutrition, body score, season, and management.

On average, and according to the present study, each Portuguese ram can produce about 20 straws for cryopreservation with 200 × 10^6^ SPZ in each [26]. Overall, the freeze-thawing cycle reduced 28.5% of the viability (alive SPZ), 35.5% of IM and increased 54.5% the SPZ defects (Figure 1). A similar decrease of SPZ viability after thawing (27.7%) was observed in three Slovak ram breeds (CASA SPZ evaluation; Slovak Dairy, Native Wallachian, and improved Wallachia rams) [27]. In our study, the breed × thawing interaction effect on SPZ viability (*p* < 0.001) highlighted differences according to the breed (Appendix A). A 36.1% decrease between fresh and thawed semen was observed in Serra da Estrela rams, which is one of the breeds with higher SPZ viability in the fresh semen (71.2%). This important breed produces the oldest traditional cheese manufactured in Portugal, Serra da Estrela Cheese (PDO) [28]. Nevertheless, the alive SPZ (45.5%) after thawing for this breed, remains similar to all the others breeds. Further research is needed to identify the causes of this high variation of SPZ viability in this breed after the freeze-thawing cycle. However, we have showed that SPZ cryopreservation is the main factor for a significant decrease in semen quality, which is normal and is in accordance to literature [27,28,29]. However, some Portuguese breeds have got good IM like some improved European breeds that trade semen doses at good prices [28].

Vozaf et al. [27] reported an IM decrease of 15.1–18.4% in their study, but a variation of 15 to 28% was observed in the Merino ram (Spain) according to three freezing methodologies [30], and up to 49.1% in Manchega rams (Spain) [31]. In a recent review considering more than 20 ram breeds, the IM decrease in thawed SPZ varied between 22.7% and 74.6% [32]. Probably, breed differences in seminal traits of frozen semen may be explained by inherent susceptibilities of SPZ cells to the deleterious effects of cryopreservation and evaluation methods used in the different studies [33]. In thawed semen, only 20–30% of SPZ cells are functional even with 40–60% motile cells [34,35].

To remain functional, SPZ should preserve their size, shape, and lipid characteristics after the freeze-thawing cycle [29]. Cryo-injuries caused by temperature stress are induced by alterations in the lipidic phase change and functional state of SPZ membranes (plasma, mitochondrial and the acrosomal membranes), osmotic stress (imbalance) created by the increasing concentration of solutes in the liquid water fraction from crystalized pure water, and oxidative stress originated by the increase of radical oxygen species [31,33]. Certainly, these are the main causes of (ultra)structural and functionally alterations, which fragilize or kill SPZ [36]. In our study, the breed × thawing interactions on SPZ traits related to SPZ defects might be due to differences in fresh seminal traits that were observed among five local breeds (Serra da Estrela, Churro Galego Mirandês, Saloio, Bordaleiro Entre Douro e Minho, and Merino da Beira Baixa) (Table 5). In thawed semen, differences for abnormal morphologies were only detected between Bordaleiro Entre Douro e Minho and Churro Galego Bragançano (Table 6). Therefore, breed differences in seminal traits in fresh semen suggest a non-meaningful impact in frozen semen of the present research. Previous work with Merino and Serra da Estrela have shown a seasonal effect of semen collection in IM, live, and normal SPZ, in frozen semen, with higher results in autumn, but suggesting that SPZ cryopreservation should be avoided in winter [9]. This aspect is relevant to ensure genetical diversity, especially for endangered sheep breeds in which the number of adult males are low [37,38]. Other researchers had shown that season of semen collection is one of the main factors that affects frozen semen quality, especially in seasonal breeds and or those exploited in higher latitudes [2,29]. Furthermore, in Merino breeds exploited in Australia (Latitude 36, South), some authors have shown a significant decrease in frozen semen quality in winter and spring [27,29]. In breeding centers placed in higher latitudes, namely in France and Ireland, photoperiod treatments associated or not with melatonin implants are routinely used during winter and spring to produce frozen semen with quality throughout the year [39,40]. Reproductive parameters have low to medium heritability varying between 0.077 to 0.304 with moderate repeatability (0.41–0.52) [25,27]. However selection based on reproduction and genetic studies should proceed to increase the number of “good freezers” to be used in artificial reproductive technologies [27,31].

In the present study, abnormal morphology of thawed SPZ increased about 55%, from 11% in fresh semen up to 17% (Figure 1c) due to the increase of head D (66.7%) and tail D (100%), while the IPD decreased 28.6%. The abnormal morphology value is considered below the threshold of 20% for thawed semen [3,27]. The crystalized water is the major cause of morphological SPZ alterations in humans [41] and domestic animals [33]. According to transmission microscopic evaluation, the cryo-damage of the SPZ head is predominant in goats and significant detachment of the SPZ head and tail occurs [42]. According to the breed × thawing interactions observed in Appendix A for the SPZ abnormal morphology (Churro do Campo rams), Appendix A for SPZ head D (Churro do Campo, Churro Galego Bragançano and Churro Algarvio rams), Appendix A for SPZ IPD (Churro do Campo rams) and Appendix A for SPZ tails D (Churro do Campo rams), a low variation of SPZ defects (%) can occurs in some of these breeds after the freeze-thawing cycle. These results also suggest that a phylogenetic association within Churra breeds for SPZ defects resilience can exist. A research line considering an omics approach to this issue may provide accurate information with impact on semen cryopreservation of rams [29,31].

The season only affected the IPD (*p* < 0.001) due to the highest percentage observed for fresh semen in period 2, which is considered the seasonal anestrus period (Figure 2), while in thawed semen, no seasonal effects were seen in IP defects. Nevertheless, a strong (*p* < 0.001) breed × season interaction on this trait was also observed, imputed not exclusively by high percentages of IPD during the breeding season (period 1; September–January) and closely to the period 2 (Appendix A) in Bordaleiro entre Douro e Minho, Churro Algarvio and Churro Galego Mirandês rams. In this work, double IP, cytoplasmatic droplets and head detachment were considered as SPZ IPD. Several studies have confirmed that sometimes it is difficult to identify which factors mainly affects frozen semen quality [43,44]. In fact, there are several factors inherent to the animal and many environmental factors as well, such as temperature, management or different SPZ cryopreservation protocols [27,29,33]. Moreover, individual variation affecting in in vivo and in vitro fertility due to different SPZ membrane composition and other inherent genetic and physiologic characteristics have been reported [29,33]. In the present study, seminal values in fresh and frozen semen are, in general, similar to results obtained in native breeds of South European countries [3,31,45], which was expected due to similarities among Mediterranean breeds and environmental conditions. We believe that it is very important to proceed with studies in Portuguese and other Mediterranean native breeds because they are very well adapted to the environmental conditions and contain unique genetic diversity that should be preserved [3,46]. Furthermore, it is of critical importance to deepen our knowledge of their reproductive potential and increase their reproductive efficiency, which is influenced by semen quality [31]. Some authors have revealed that reproductive parameters are correlated with ejaculate traits namely mass motility, concentration, and post thaw IM [3,29].

## 5. Conclusions

We conclude, despite breed differences in fresh and frozen semen, all of them were able to produce semen throughout the whole year for mating, cryopreservation purposes, and assisted reproductive biotechnologies, without the use of photoperiodic or hormonal (melatonin implants) treatments. The effect of season on these reproductive traits is low in both semen types and restricted to some breeds, except for IPD. We noticed an individual variation of seminal parameters in both semen types.

## Figures and Tables

**Figure 1 animals-13-00579-f001:**
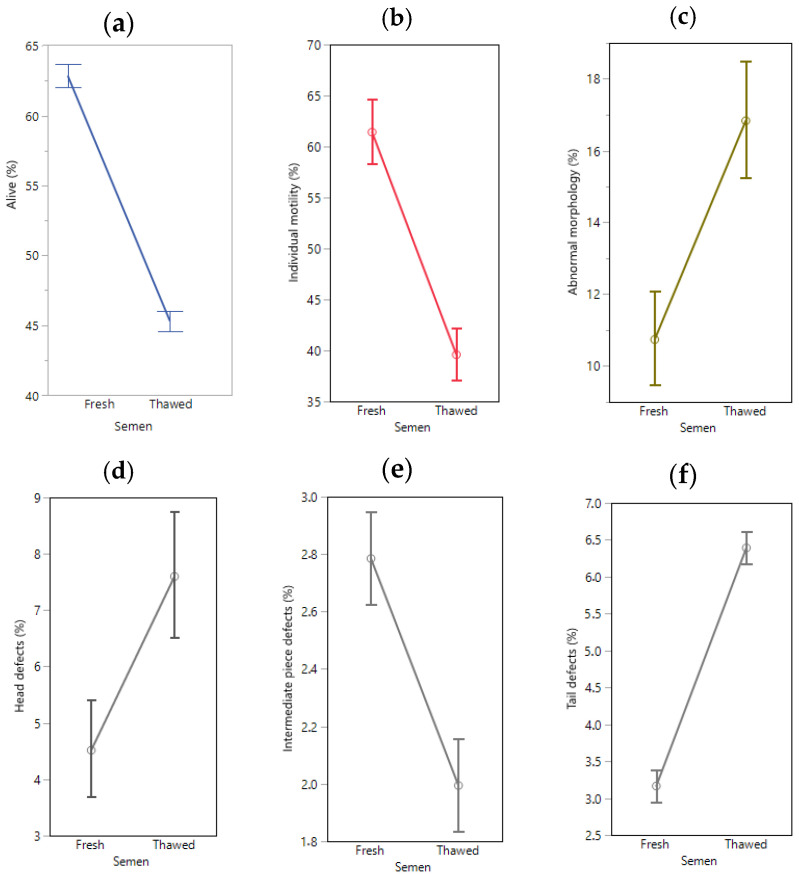
LSmean differences (*p* < 0.001) of spermatozoa alive (**a**), individual motility (**b**), abnormal morphology (**c**), head (**d**), intermediate piece (**e**) and tail (**f**) defects in fresh and thawed semen for 10 Portuguese local ram breeds. The vertical bars represent the 95% confidence interval.

**Figure 2 animals-13-00579-f002:**
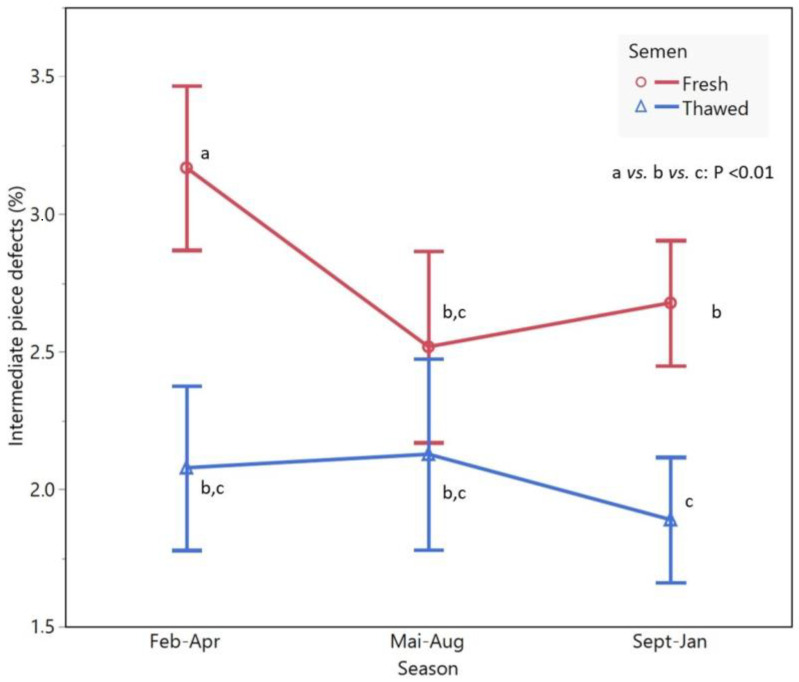
LSmean differences (*p* < 0.01) of intermediate piece by the season and processing semen. The vertical bars represent the 95% confidence interval.

**Table 1 animals-13-00579-t001:** Ram breeds, number of animals and ejaculates obtained from 2004 till 2020.

Breed	Number of Rams	Number of Ejaculates
Serra da Estrela	22	224
Churro Galego Mirandês	12	224
Saloio	11	235
Bordaleiro Entre Douro e Minho	10	253
Churro Galego Bragançano	7	176
Churro do Campo	7	51
Merino Beira Baixa	6	121
Mondegueira	5	97
Churro Algarvio	3	49
Merino Branco	2	41
Total	85	1471

**Table 2 animals-13-00579-t002:** Descriptive statistics of ejaculate traits for each breed.

Breed	SPZ Traits								
	Volume per Ejaculate (mL)			SPZ Concentration (10^6^/mL)			Total per Ejaculate (10^6^)		
	$\tilde{x}$	$\bar{x}$	CV%	$\tilde{x}$	$\bar{x}$	CV%	$\tilde{x}$	$\bar{x}$	CV%
Serra da Estrela	0.80	0.83	41	5100	5294	44	3918	4377	56
Churro Galego Mirandês	0.70	0.72	33	5420	5644	34	3714	4181	53
Saloio	0.90	0.90	34	4880	5009	31	4160	4571	49
Bordaleiro Entre Douro e Minho	0.50	0.56	47	5200	5470	47	2840	3114	60
Churro Galego Bragançano	0.90	0.88	41	4100	4042	31	3658	3752	57
Churro do Campo	0.50	0.53	31	3650	3660	35	1835	2041	59
Merino Beira Baixa	0.65	0.65	40	5200	5422	36	3050	3589	56
Mondegueira	0.85	0.88	80	4000	4060	27	3180	3516	65
Churro Algarvio	0.90	0.92	33	4740	4491	29	4320	4217	46
Merino Branco	0.85	0.85	31	6000	5670	21	4320	4928	41
Total	0.75	0.77	49	4900	5039	38	3523	3894	57

**Table 3 animals-13-00579-t003:** Ejaculate volume, spermatozoa concentration, and total sperm per ejaculate in the 10 local ram breeds (*n* = 1471).

Breed		SPZ Traits *	
	Volume per Ejaculate (mL)	SPZ Concentration (10^6^/mL)	Total per Ejaculate (10^6^)
Serra da Estrela	0.84 ± 0.02 ^a^	5283 ± 66 ^a,b^	4396 ± 79 ^a^
Churro Galego Mirandês	0.72 ± 0.03 ^a,b,c^	5916 ± 79 ^a^	4319 ± 93 ^a^
Saloio	0.88 ± 0.03 ^a^	4889 ± 79 ^a,b^	4315 ± 93 ^a^
Bordaleiro Entre Douro e Minho	0.54 ± 0.02 ^c^	5.303 ± 83 ^a,b^	2881 ± 100 ^a,b^
Churro Galego Bragançano	0.81 ± 0.03 ^a,b^	3879 ± 98 ^b^	3248 ± 115 ^a,b^
Churro do Campo	0.52 ± 0.05 ^b,c^	3293 ± 136 ^b^	1764 ± 169 ^b^
Merino Beira Baixa	0.58 ± 0.04 ^b,c^	5068 ± 125 ^a,b^	2990 ± 149 ^a,b^
Mondegueira	0.83 ± 0.04 ^a,b^	3959 ± 125 ^a,b^	3291 ± 147 ^a,b^
Churro Algarvio	0.87 ± 0.05 ^a,b^	4011 ± 159 ^a,b^	3581 ± 188 ^a,b^
Merino Branco	0.85 ± 0.06 ^a,b,c^	5423 ± 197 ^a,b^	4718 ± 234 ^a,b^

* LSmean ± (SQRT)SEM; SPZ: Spermatozoa; ^a–c^ different superscript letters for each column: *p* < 0.05.

**Table 4 animals-13-00579-t004:** Effects of breed, season, and semen cryopreservation on reproductive SPZ traits.

Spermatozoa (%)	Breed	Season	Thawing	Breed × Season	Breed × Thawing	Season × Thawing	Breed × Season × Thawing
Alive	**	NS	***	NS	***	NS	NS
Individual motility	*	NS	***	*	NS	NS	**
Abnormal morphology	**	NS	***	**	***	*	NS
Head defects	***	NS	***	***	***	NS	NS
Intermediate piece defects	*	***	***	***	*	*	NS
Tail defects	NS	NS	***	NS	**	NS	NS

* *p* < 0.05; ** *p* < 0.01; *** *p* < 0.001.

**Table 5 animals-13-00579-t005:** Effects of breed and breed × semen processing interactions on spermatozoa traits of fresh semen.

Breed	Spermatozoa Traits (%) *					
	Alive	IM	AM	Head D	IP D	Tail D
Serra da Estrela	71.0 ± 0.3 ^a^	54.3 ± 0.2	6.5 ± 0.2 ^a^	2.7 ± 0.2 ^a,c^	1.0 ± 0.1 ^a^	1.6 ± 0.1
Churro Galego Mirandês	61.5 ± 0.4 ^b^	67.0 ± 0.2	11.9 ± 0.2 ^b^	4.7 ± 0.2 ^a,b,c^	2.3 ± 0.1 ^b^	2.5 ± 0.1
Saloio	63.0 ± 0.4 ^b^	63.4 ± 0.2	9.5 ± 0.2 ^a,b^	4.1 ± 0.2 ^a,b,c^	1.5 ± 0.1 ^a,b^	2.2 ± 0.1
Bordaleiro Entre Douro e Minho	63.4 ± 0.4 ^a,b^	63.9 ± 0.2	15.1 ± 0.2 ^b^	7.9 ± 0.2 ^b^	2.4 ± 0.1 ^b^	2.8 ± 0.1
Churro Galego Bragançano	63.5 ± 0.5 ^a,b^	59.3 ± 0.3	8.5 ± 0.3 ^a,b^	2.4 ± 0.3 ^c^	1.9 ± 0.1 ^a,b^	2.8 ± 0.1
Churro do Campo	54.2 ± 0.6 ^b^	54.7 ± 0.3	11.7 ± 0.4 ^a,b^	5.5 ± 0.4 ^a,b,c^	0.8 ± 0.3 ^a,b^	3.7 ± 0.3
Merino Beira Baixa	74.6 ± 0.6 ^a^	67.2 ± 0.3	13.8 ± 0.3 ^b^	6.4 ± 0.3 ^a,b,c^	2.2 ± 0.2 ^a,b^	2.5 ± 0.2
Mondegueira	60.7 ± 0.6 ^a,b^	63.9 ± 0.3	11.1 ± 0.3 ^a,b^	6.9 ± 0.3 ^a,b,c^	1.9 ± 0.2 ^a,b^	2.0 ± 0.2
Churro Algarvio	64.1 ± 0.7 ^a,b^	59.7 ± 0.4	11.7 ± 0.4 ^a,b^	2.7 ± 0.4 ^a,b,c^	2.1 ± 0.2 ^a,b^	2.1 ± 0.2
Merino Branco	59.2 ± 0.9 ^a,b^	61.6 ± 0.5	11.5 ± 0.5 ^a,b^	3.7 ± 0.5 ^a,b,c^	2.4 ± 0.2 ^a,b^	3.1 ± 0.3

^a–c^ different superscript letters for each column: *p* < 0.05. * LSmean ± (Sqrt)SEM. IM: individual motility; AM: Abnormal morphology; IP: Intermediate piece; D: Defects.

**Table 6 animals-13-00579-t006:** Effects of breed and breed × semen processing interactions on spermatozoa traits of thawed semen.

Breed	Spermatozoa Traits (%)					
	Alive	IM	AM	Head D	IP D	Tail D
Serra da Estrela	45.5 ± 0.3	34.1 ± 0.2	14.5 ± 0.2 ^a,b^	7.4 ± 0.2 ^a,b,c^	1.0 ± 0.1	4.4 ± 0.1
Churro Galego Mirandês	50.8 ± 0.4	44.9 ± 0.2	19.2 ± 0.2 ^a,b^	9.9 ± 0.2 ^a,c^	1.1 ± 0.1	5.2 ± 0.1
Saloio	45.6 ± 0.4	40.6 ± 0.2	14.9 ± 0.2 ^a,b^	6.6 ± 0.2 ^a,b,c^	1.5 ± 0.1	5.6 ± 0.1
Bordaleiro Entre Douro e Minho	48.3 ± 0.4	42.1 ± 0.2	22.6 ± 0.2 ^a^	13.2 ± 0.2 ^a^	1.1 ± 0.1	5.6 ± 0.1
Churro Galego Bragançano	43.0 ± 0.5	38.6 ± 0.3	12.2 ± 0.3 ^b^	2.8 ± 0.3 ^b^	1.1 ± 0.1	6.7 ± 0.1
Churro do Campo	34.6 ± 0.6	28.8 ± 0.3	13.2 ± 0.4 ^a,b^	7.0 ± 0.4 ^a,b,c^	1.1 ± 0.3	4.0 ± 0.3
Merino Beira Baixa	53.4 ± 0.6	46.0 ± 0.3	24.1 ± 0.3 ^a,b^	12.1 ± 0.3 ^a,b,c^	1.5 ± 0.2	7.4 ± 0.2
Mondegueira	48.7 ± 0.6	42.7 ± 0.3	19.3 ± 0.3 ^a,b^	9.7 ± 0.3 ^a,b,c^	1.2 ± 0.2	6.5 ± 0.2
Churro Algarvio	38.8 ± 0.7	38.5 ± 0.4	13.0 ± 0.4 ^a,b^	3.1 ± 0.4 ^c^	1.4 ± 0.2	6.6 ± 0.2
Merino Branco	43.4 ± 0.9	41.3 ± 0.5	17.6 ± 0.5 ^a,b^	7.8 ± 0.5 ^a,b,c^	1.5 ± 0.2	7.4 ± 0.3

^a–c^ different superscript letters for each column: *p* < 0.05. IM: individual motility; AM: Abnormal morphology; IP: Intermediate piece; D: Defects.

## Data Availability

The data present in this study are available on request from the corresponding author.

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
