# Peer review of "Ram Semen Cryopreservation for Portuguese Native Breeds: Season and Breed Effects on Semen Quality Variation"

_animals, 2023, doi:10.3390/ani13040579_

Round 1

Reviewer 1 Report

Line 66: Remove one dot.

Line 76: Add dot after temperatures word.

Line 121: remove till and add: to

Lines 131, 151,153, 165, 168, 340: Remove Symbol (º)

Line 374: in vivo and in vitro (cursive).

¿Animal age?

Lines 256, 364: Remove Fev-Apr (February- April)

Table 5. Check the superscript letters.

In alive sperm (%), Serra da estela (38.2+0.3) was similar to churro Galego Bragançano (63.5+0.5), however, Mirandes (61.5+0.4), Saloio (63.0±0.4), Bordaleiro Entre Douro e Minho (63.4±0.4) were differents to Serra,

¿why sperm viability showed a low percentage (8-10%) between fresh and thawed semen samples for these local ram breeds?

As a generalisation, some 40-50% of the sperm population does not survive cryopreservation even with optimised protocols.

It´s widely known that cryopreservation of sperm cells has deleterious effects in the plasma membrane integrity. Likewise, ram sperm cells are more sensitive to extreme temperature changes during the freezing process and suffer more damage than different animal species. This is due to several factors caused by cryopreservation protocols that alter sperm integrity, such as formation/reshaping of intracellular ice during freezing and thawing and dissolution of ice in the extracellular environment. All of these factors induce sperm membrane and tail damage.

Author Response

Line 66: Remove one dot.

 Done

Line 76: Add dot after temperatures word.

Done

Line 121: remove till and add: to

 Done

Lines 131, 151,153, 165, 168, 340: Remove Symbol (º)

Done

Line 374: in vivo and in vitro (cursive).

Done

¿Animal age?

Done

Lines 256, 364: Remove Fev-Apr (February- April):

Done

Table 5. Check the superscript letters.

In alive sperm (%), Serra da estela (38.2+0.3) was similar to churro Galego Bragançano (63.5+0.5), however, Mirandes (61.5+0.4), Saloio (63.0±0.4), Bordaleiro Entre Douro e Minho (63.4±0.4) were differents to Serra,

A writing error of % alive SPZ in fresh and thawed SPZ of Serra da Estrela was detected and corrected, as well some superscripts. Thanks! For the remaining traits all is right. Note that the SEM, which is the SQRT transformation, is different for some breeds and this dispersion influenced the statistical significance. 

¿why sperm viability showed a low percentage (8-10%) between fresh and thawed semen samples for these local ram breeds?

As a generalisation, some 40-50% of the sperm population does not survive cryopreservation even with optimised protocols.

It´s widely known that cryopreservation of sperm cells has deleterious effects in the plasma membrane integrity. Likewise, ram sperm cells are more sensitive to extreme temperature changes during the freezing process and suffer more damage than different animal species. This is due to several factors caused by cryopreservation protocols that alter sperm integrity, such as formation/reshaping of intracellular ice during freezing and thawing and dissolution of ice in the extracellular environment. All of these factors induce sperm membrane and tail damage.

In overall, the decrease was 28.5% after correcting Serra da Estrela values (L307). In original version, the decrease of 8-10% in Serra da Estrela was due to this mistake. Thanks for your advice.

Reviewer 2 Report

This study mainly explores the effects of different species and different seasons on spermatozoa. The amount of data is rich and the article is rigorous, but some revisions are needed.

Lines 57-68: Is there any necessary connection between the before and after presentation? It is suggested to add references to increase the readability of the article.

Lines 378-346: There are few references to the discussion of the results in the article, and [25] and [27] appear to be cited several times, can they be summarized together?

It is suggested that new references be introduced in the discussion section for corroboration.

Author Response

Comments and Suggestions for Authors

This study mainly explores the effects of different species and different seasons on spermatozoa. The amount of data is rich and the article is rigorous, but some revisions are needed.

Lines 57-68: Is there any necessary connection between the before and after presentation? It is suggested to add references to increase the readability of the article.

Done. New (10) appropriate references were added in this version.

Lines 378-346: There are few references to the discussion of the results in the article, and [25] and [27] appear to be cited several times, can they be summarized together?

Done. The citation management is improved to increase the readability of the discussion, and according the main results of this study

It is suggested that new references be introduced in the discussion section for corroboration.

Done